# Unveiling the Molecular Landscape of *FOXA1* Mutant Prostate Cancer: Insights and Prospects for Targeted Therapeutic Strategies

**DOI:** 10.3390/ijms242115823

**Published:** 2023-10-31

**Authors:** Kyung Won Hwang, Jae Won Yun, Hong Sook Kim

**Affiliations:** 1Department of Biological Sciences, Sungkyunkwan University, Suwon 16419, Republic of Korea; hgw7656@gmail.com; 2Veterans Health Service Medical Research Institute, Veterans Health Service Medical Center, Seoul 05368, Republic of Korea; jwyunmd@gmail.com

**Keywords:** *FOXA1*, prostate cancer, drug repurposing, gene expression, pathway analysis, gene–drug network

## Abstract

Prostate cancer continues to pose a global health challenge as one of the most prevalent malignancies. Mutations of the Forkhead box A1 (*FOXA1*) gene have been linked to unique oncogenic features in prostate cancer. In this study, we aimed to unravel the intricate molecular characteristics of *FOXA1* mutant prostate cancer through comprehensive in silico analysis of transcriptomic data from The Cancer Genome Atlas (TCGA). A comparison between *FOXA1* mutant and control groups unearthed 1525 differentially expressed genes (DEGs), which map to eight intrinsic and six extrinsic signaling pathways. Interestingly, the majority of intrinsic pathways, but not extrinsic pathways, were validated using RNA-seq data of 22Rv1 cells from the GEO123619 dataset, suggesting complex biology in the tumor microenvironment. As a result of our in silico research, we identified novel therapeutic targets and potential drug candidates for *FOXA1* mutant prostate cancer. KDM1A, MAOA, PDGFB, and HSP90AB1 emerged as druggable candidate targets, as we found that they have approved drugs throughout the drug database CADDIE. Notably, as most of the approved drugs targeting MAOA and KDM1A were monoamine inhibitors used for mental illness or diabetes, we suggest they have a potential to cure *FOXA1* mutant primary prostate cancer without lethal side effects.

## 1. Introduction

Prostate cancer is one of the most diagnosed cancers in men, and despite significant advances in diagnosis and treatment, it remains a major cause of mortality from cancer in men globally [1]. One of the major challenges in managing prostate cancer is the heterogeneity of the disease driven by genetic alterations such as gene mutations and chromosomal rearrangement, leading to diverse clinical outcomes and responses to treatment [2]. Recent advances in genomic and molecular profiling technologies have led to a deeper insight of the mechanisms underlying prostate cancer development and progression, as well as the responses to various drugs [3,4,5]. It emphasizes the importance of precision medicine and targeted therapies that aim to tailor medical treatment to an individual’s unique characteristics, which offers more effective and less toxic treatments than traditional chemotherapy [6,7,8].

The molecular landscape of prostate cancer is characterized by recurrent genomic alterations, including the fusion of ETS family genes such as *ERG*, *ETV1*, *ETV4*, or *FLI1*, and genetic mutations involving *SPOP*, *FOXA1*, or *IDH1* [4]. *FOXA1* (Forkhead box protein A1), in particular, is frequently mutated, occurring in up to 9% and 13% of cases of primary prostate cancer and metastatic castration-resistant prostate cancer (mCRPC), respectively [3,9]. In the context of neuroendocrine prostate cancer (NEPC), *FOXA1* mutations are observed in approximately 25% of tumors [10]. Interestingly, the frequency of *FOXA1* mutation varies with ethnic background, with a striking finding of *FOXA1* mutations in 41% of localized prostate tumors within the Chinese Prostate Cancer Genome and Epigenome Atlas (CPGEA), which is significantly higher than western cohorts [11]. While *FOXA1* mutation can affect various regions of its coding sequence, more than 50% of these mutations affect the nucleotides encoding the Forkhead DNA binding domain, thereby impacting FOXA1 function as a pioneer factor [4,11,12,13].

FOXA1 plays a pivotal role in promoting accessible chromatin conformations, thereby enabling the appropriate binding of specific transcription factors such as AR to the genome in prostate tissue [14]. This regulatory mechanism orchestrated by FOXA1 is pertinent to the transcriptional programs observed in both normal and cancerous prostate tissue [15,16,17]. Meanwhile, mutation of *FOXA1* alters the tumor environment significantly so that it is regarded as a driver gene of prostate cancer [3,4]. Although numerous reports have demonstrated the role of *FOXA1* in both in vitro and in vivo prostate cancer models, the precise role and impact of *FOXA1* mutations in primary prostate cancer and tumor microenvironment remain to be fully elucidated. Limitations in its clinical application also remain.

In this study, we aimed to explore the molecular and cellular characteristics associated with *FOXA1* mutant prostate cancer and identify potential therapeutic targets and candidate drugs that may hold promise for improved treatment approaches. We conducted an analysis of TCGA primary prostate adenocarcinoma (PRAD) mRNA-seq data to compare two groups: a *FOXA1* mutant group and a mutation-negative (control) group. Our investigation focused on identifying significant genes and pathways related to tumorigenesis, tumor development, and the immune system. Moreover, we expected that the bioinformatic tools and analytical pipeline used in this paper could be valuable for investigating the roles of different genes in various cancer types.

## 2. Results

### 2.1. Patient Selection for the FOXA1 Mutant Group and Control Group

The *FOXA1* mutant and mutant-negative (control) groups for prostate cancer were selected using genomic data [4]. To reduce false positive results caused by the difference in number between the two groups, the control group was further defined by considering total mutation counts, *FOXA1* expression, and *AR* expression levels. The patient characteristics are presented in Table 1, and the study flowchart is provided in Figure 1. The Gleason sum exhibited significant differences between the two groups, consistent with prior research indicating that the *FOXA1* mutation has an impact on higher Gleason scores (Table 1) [18,19]. However, factors such as age, race, and primary tumor laterality did not reveal significant differences (Table 1). Moreover, both overall survival and disease-free survival did not exhibit any distinctions between the groups (Table 1 and Appendix A).

Among the nine mutant samples, seven were located in the Forkhead domain, and the other two were located near the C-terminal part of the *FOXA1* gene (Appendix A). Four samples had single nucleotide polymorphism (SNP), two had deletion, one had the insertion type of *FOXA1* mutation, and another one had both SNP and deletion (Appendix A). Heterogeneous effects derived from each *FOXA1* mutation were not expected, as most of the mutations were found within the Forkhead domain.

### 2.2. Molecular Characteristics in the FOXA1 Mutant Group and Control Group

To understand which genes are significantly related to *FOXA1* mutation, we performed differentially expressed genes (DEGs) analysis with the Student’s *t*-test and obtained the *p*-value of each gene. When selecting genes with a *p*-value of less than 0.05, we obtained 1525 DEGs that were significantly associated with *FOXA1* mutations (Appendix A). We performed pathway analysis using ConsensusPathDB (CPDB) with these genes to identify which cellular signaling pathways were significantly related. We found that eight cancer-related pathways—TP53, Hippo signaling pathway, WNT signaling pathway, GPCR signaling pathway, Notch signaling pathway, ERBB2 and ERBB4 signaling pathways, MAPK6/MAPK4 signaling pathway, and cell adhesion molecules—were markedly altered in the *FOXA1* mutant group, and 132 genes out of 1525 genes were included in these pathways (Figure 2A,B and Appendix A). In particular, Notch signaling was derived from upregulated DEGs, and TP53, GPCR, ERBB signaling was from downregulated DEGs in the *FOXA1* mutant group (Appendix A). Furthermore, 19 genes contributed to multiple pathways, highlighting the importance of considering these altered pathways and the genes involved in them (Figure 2C). *FZD7*, *FZD8*, *PAK1*, *PIK3R1*, *WNT1*, *WNT2B*, and *WNT7B* were present in three different pathways, while other genes were present in two different pathways. These genes mostly included Hippo signaling, WNT signaling, and/or GPCR signaling. We validated our pathway analysis results with the previously published GEO123619 dataset [12]. Among the eight cancer-related pathways, excluding GPCR signaling, seven pathways exhibited significant alterations in the GSE123619 dataset, which reinforces the validity of our analysis (Figure 2D and Appendix A).

FOXA1 as a pioneer factor plays an important role in recruiting AR into a specific region in a genome by changing chromatin conformation and leads to AR-dependent gene expressions [20]. Thus, we examined whether AR signaling and related gene expression was altered by *FOXA1* mutation by performing GSEA analysis and comparing the expression of 182 AR signaling-associated genes. The result revealed that there was no significant difference between the *FOXA1* mutant and the control group in AR signaling and related gene expression (Figure 2E, Appendix A).

### 2.3. Tumor-Immune Phenotypes in the FOXA1 Mutant Group and Control Group

Tumor-immune phenotypes related to *FOXA1* were studied by focusing on six immune-related pathways, including CD28 co-stimulation, IL2 signaling, cell recruitment, cytokine–cytokine receptor interaction, interferon signaling, and IL12 signaling (Figure 3A). These pathways involved 56 genes with *p*-values of less than 0.05. Most of these genes were highly expressed in the control group compared to *FOXA1* mutant prostate cancer (Figure 3B and Appendix A). Furthermore, the expression values of some genes—related to chemokines and cytokines and showing *p* < 0.05—were visualized for easy comparison (Figure 3C). To extend the analysis, the six immune-related pathways were explored using the GSE123619 dataset, which pertains to prostate cancer 22Rv1 cells [12]. Intriguingly, only two out of the six immune-related pathways, namely IL2 signaling and interferon signaling, were found to overlap between the TCGA RNA-seq data and the GSE123619 dataset (Figure 3D and Appendix A). It is important to note that within tumor tissue, cancer cells engage in complex interactions with surrounding cells, particularly immune cells. With this point of view, the TCGA RNA-seq data are expected to capture more active immune-related pathways [21].

Furthermore, immune cell features in a tumor microenvironment (TME) were investigated using multiple bulk RNA-seq deconvolution tools. B cells were significantly highly present in the control group compared to *FOXA1* mutant prostate cancer using TIMER (Figure 4A). Additionally, GSEA analysis revealed that negative regulation of B cell activation and abnormal B cell morphology were upregulated in the *FOXA1* mutant group (Appendix A), providing further support for the differences in B cell-related processes between the two groups. TIP algorithms provided immune activity scores in each step in anticancer immunity [22]. Interestingly, regulatory T cell recruiting in Step 4 of the cancer–immunity cycle was lower in *FOXA1* mutant prostate cancer compared to the control group (Figure 4B).

### 2.4. Selection of FOXA1-Associated Cancer Genes Based on the CancerMine Database

Next, we conducted analysis of cancer-related genes within a dataset of 1525 genes using the CancerMine database (Figure 5A). We determined upregulated or downregulated genes by calculating the fold change (FC) between the *FOXA1* mutant group and the control group. We identified 168 genes as upregulated genes and 1357 genes as downregulated genes in the mutant group. Subsequently, we aligned these upregulated or downregulated genes with a pool of 6585 cancer genes from the CancerMine database. As a result, 58 cancer genes were upregulated and 436 cancer genes were downregulated in *FOXA1* mutant prostate cancer (Figure 5A). Within these 6585 cancer genes, there were 726 oncogenes/driver genes and 482 tumor suppressor genes associated with prostate cancer. In *FOXA1* mutant prostate cancer, 9 of the upregulated genes were oncogenes/driver genes, while 45 of the downregulated genes were tumor suppressor genes specifically related to prostate cancer (Figure 5B).

### 2.5. Identification of Potential Therapeutic Targets and Targeted Drugs for FOXA1 Mutant Prostate Cancer Patients

Nine of the upregulated genes (*CD81*, *EIF4A1*, *KDM1A*, *MAOA*, *METTL3*, *PDGFB*, *STEAP1*, *STEAP2*, and *TRIM28*) identified in *FOXA1* mutant prostate cancer, which also overlapped with known oncogenes/driver genes in prostate cancer, were subjected to further exploration of potential therapeutic options. We conducted further analysis using the gene–drug network database, CADDIE, with a specific focus on approved drugs and complex compound drugs, excluding simple chemicals. PDGFB, KDM1A, and MAOA were identified as potential therapeutic targets, with approved candidate drugs. Specifically, for PDGFB, we found three suggested approved drugs: Sunitinib, Afatinib, and Axitinib. KDM1A was associated with seven drugs, and MAOA with six drugs, many of which were monoamine oxidase inhibitors prescribed for depression and hypertension or PPAR stimulators used in diabetes treatment. This finding was further supported by the significant alterations in monoamine neurotransmitter-associated pathways resulting from ORA (Appendix A).

In addition, we conducted a further exploration of potential drug targets from a different perspective, with protein–protein interaction (PPI) analysis. We input *FOXA1* and the 168 upregulated DEGs to identify physical protein interactions among the overexpressed genes in *FOXA1* mutant prostate cancer via the PPI analysis tool STRING [23]. As the results displayed connected nodes representing proteins encoded by the 168 upregulated genes and other proteins mediating their interactions, six proteins (HSP90AB1, KDM1A, FKBP4, H2BC15, WNT7B, and GRB7) were identified as direct interactors with FOXA1 (Appendix A). Subsequently, we explored the DepMap database using these six proteins. Compound Viability data with a particular focus on the PRISM Repurposing Public 23Q2 dataset were explored, as this provided drug response data with associated protein targets [24]. Among the six proteins, we found experimental data for HSP90AB1 when investigating prostate cancer cell lines. HSP90AB1 was targeted by HSP inhibitor NVP-AUY922 (Luminespib) in 22Rv1, PC3, DU145, and LNCaP. However, this drug has not yet been approved for use in patients, although previous studies have reported its anticancer effect in animal models of endometrial cancers, leukemia, and melanoma [25,26,27]. Additionally, we input the genes encoding these proteins into CADDIE, excluding simple chemicals and antibiotics. Subsequently, HSP90AB1 was targeted by two approved candidate drugs, Diacerein and Dipyridamole. For each drug, we provide detailed information on the mode of action (MOA) and the approved diseases alongside their respective targets (Table 2).

## 3. Discussion

Prostate cancer is a heterogeneous disease with diverse molecular alterations contributing to its development and progression. Among the key genetic alterations observed in prostate cancer, mutations in the *FOXA1* gene have been found in up to 9% of primary prostate cancer cases and have gained considerable attention [4]. FOXA1 plays a crucial role in normal prostate development and differentiation, and its mutation has been considered as a potential driver in tumor malignancy [14,18].

In this research, we investigated the cellular and molecular characteristics of *FOXA1* mutant prostate cancer using the TCGA dataset. Initially, we grouped patients based on *FOXA1* mutation status, resulting in 9 patients with *FOXA1* mutant and 86 patients with no driver mutations. Subsequently, we considered additional factors, total mutation counts, *FOXA1* expression levels, and *AR* expression levels (Table 1). By applying these criteria, we further refined the control group to include 46 patients, thereby minimizing potential confounding factors and enhancing the accuracy of our analysis.

First, gene expressions altered by *FOXA1* mutation were analyzed by comparing gene expression levels between two groups, and 1525 DEGs were obtained. These genes were utilized for the pathway analysis, and eight cancer-intrinsic signaling pathways (TP53, Hippo signaling pathway, WNT signaling pathway, GPCR signaling pathway, Notch signaling pathway, ERBB2 and ERBB4 signaling pathways, MAPK6/MAPK4 signaling pathway, and cell adhesion molecules) and six cancer-extrinsic signaling pathways (CD28 co-stimulation, IL2 signaling, cell recruitment, cytokine–cytokine receptor interaction, interferon signaling, and IL12 signaling) were identified (Figure 2A and Figure 3A). Many genes under these cancer-intrinsic and extrinsic pathways were downregulated in the *FOXA1* mutant group compared to the control group (Figure 2B, Figure 3B and Appendix A), suggesting the limitation of the FOXA1 function as a transcription factor in *FOXA1* mutant prostate cancer. This notion was further supported by the downregulation of tumor suppressor genes within *FOXA1* mutant prostate cancer. For example, previously known WNT suppressors such as CTNNBIP1, SERPINF1(PEDF), GPC4, and SOST and tumor suppressors such as GNAO1, PRKG1, VANGL2, FZD7, FZD8, RSPO3, ITPR1, SMURF2, and MAPK8 were downregulated in the *FOXA1* mutant group.

In the context of cancer-intrinsic pathways, a notable finding emerged. Despite AR signaling being a major pathway regulated by FOXA1, our study did not uncover significant changes in AR signaling between the *FOXA1* mutant group and the control group, as revealed by pathway analysis. Our GSEA analysis using AR signaling-related gene sets also yielded no significant differences. Additionally, the expression of individual genes used for GSEA analysis did not show any significant variation between the two groups (Figure 2E and Appendix A). Notably, a prior study identified that the quantity of FOXA1 serves as a balancing factor in controlling the AR program [20]. In our study, we classified the control and *FOXA1* mutant groups based on *FOXA1* mutation status, and we additionally considered the gene expression of *AR* and *FOXA1* (Table 1). This resulted in similar *FOXA1* levels across both groups, thereby explaining the lack of divergence in AR signaling.

In terms of cancer-extrinsic pathways, tumors comprise various cell types including cancer cells, normal cells, immune cells, and fibroblasts with intricate communication among them crucial for tumor fate. Despite the widely accepted recognition of the importance of tumor microenvironment, comprehensive investigation into specific aspects such as immune-related signaling in *FOXA1* mutant prostate cancer has remained largely unexplored. Intriguingly, our study has unveiled a significant finding, with genes within six immune-related pathways generally displaying lower expression in the *FOXA1* mutant group compared to the control group, with downregulation of chemokines and cytokines (Figure 3B,C). This observation gains further support from clinical findings. It is noteworthy that the mutation ratio of *FOXA1* in castration-resistant prostate cancer (CRPC) is significantly higher compared to primary prostate cancer [12]. CPRC has exhibited limited responsiveness to immune checkpoint inhibitors, attributed to its inherently immunosuppressive characteristics [28]. Specifically, we observed the downregulation of genes involved in the IFN signaling pathway, which is a crucial component for the response to immune checkpoint inhibitors. This points to the presence of an immune-suppressive mechanism associated with *FOXA1* mutant cells. Digging deeper into immune-related pathways, we identified six pathways associated with immune inactivation in *FOXA1* mutant prostate cancer. Notably, the CD28 co-stimulatory pathway, IL2 signaling, and IL12 signaling are all intricately linked to T cell activation, including the production of cytokines like IFN-γ upon antigen recognition [29,30,31,32]. Interestingly, a previous study reported a negative correlation between *FOXA1* expression and interferon signatures and antigen presentation gene expression, regardless of the presence or absence of *FOXA1* mutation [33]. While our findings appear to diverge from this previous research, a comprehensive analysis of earlier studies collectively does suggest a role for FOXA1 in immune system regulation [12,28].

To gain a deeper understanding of the immune system’s relationship with *FOXA1* mutation at a cellular level, we performed RNA-seq deconvolution analysis. Among several immune infiltration estimation tools, we selected TIMER because TIMER was initially developed based on TCGA samples [34]. Our analysis revealed that B cells are significantly enriched in the control group compared to *FOXA1* mutant prostate cancer, and this result was further confirmed by GSEA analysis (Figure 4A and Appendix A). Considering previous studies highlighting the importance of CXCL13 axis in B cell recruitment [35,36,37] and our finding of *CXCL13* downregulation in *FOXA1* mutant prostate cancer (Figure 3C and Appendix A), it is plausible to suggest that CXCL13 may be a contributing factor to these results. Moreover, our exploration of anticancer immunity by using TIP tools revealed a significant difference between the control and *FOXA1* mutant groups, particularly in step 4 involving T reg cell recruitment (Figure 4B). This finding is supported by a previous study indicating the influence of CXCL13 on T reg cell activation [38,39]. While additional validation is required to elucidate the specific impact of *FOXA1* mutation on these immune-related pathways, our findings propose that *FOXA1* mutation in prostate cancer leads to alterations in tumor-associated immune systems.

Our study was mainly conducted using TCGA cohorts. To examine whether the results were cohort dependent, we collected and analyzed another dataset from the GEO database. Interestingly, the majority of cancer-intrinsic signaling pathways, except the GPCR signaling pathway, were affirmed in this dataset, while only two out of six cancer-extrinsic signaling pathways were identified (Figure 2D and Figure 3D). Notably, the data from the GEO database were generated using an in vitro cell culture system with prostate cancer cell line, 22Rv1 [12]. As a result, the altered cancer-extrinsic signaling pathways may have been limited to those influenced solely by the cell culture medium, lacking signals from surrounding cells such as immune cells or fibroblasts, which are present in the real tumor environment. This discrepancy highlights the differences between in vitro and in vivo tumor environments, where the immune landscape is intricately orchestrated by various cells and tissues.

Based on previous reports and our current study, FOXA1 is an attractive therapeutic target. However, as of the present date, there are no targeted drugs available for *FOXA1* mutation. Considering the difficulties of drug development for transcription factor due to its structural disorder and involvement in various pathways, we have taken a different approach to identify a candidate therapeutic target for *FOXA1* mutant prostate cancer using its own molecular characteristics [40]. To begin, we aligned the DEGs, and this allowed us to match the gene expression pattern in the *FOXA1* mutant group with the list of oncogenes/driver genes and tumor suppressor genes in the CancerMine database. Consequently, we identified that nine oncogenes or tumor driver genes—*CD81*, *EIF4A1*, *KDM1A*, *MAOA*, *METTL3*, *PDGFB*, *STEAP1*, *STEAP2*, and *TRIM28*—were upregulated in *FOXA1* mutant prostate cancer (Figure 5B). Notably, previous studies have shown that CD81, METTL3, STEAP1, and TRIM28 contributed to the progression of prostate cancer [41,42,43,44]. In addition, we explored therapeutic targets by investigating FOXA1 interacting protein within proteins encoded by upregulated genes in *FOXA1* mutant prostate cancer, and identified six genes—*HSP90AB1*, *KDM1A*, *FKBP4*, *H2BC15*, *WNT7B*, and *GRB7* (Appendix A). Interestingly, among the genes, PDGFB, MAOA, KDM1A, and HSP90AB1 were targets of approved actionable drugs within the CADDIE databases.

The mode of action (MOA) and approved diseases for each drug were listed alongside their respective targets (Table 2). Among 16 candidate drugs, 6 (Sunitinib, Rosiglitazone, Phenelzine, Pioglitazone, Pargyline, and Dipyridamole) were reported for their anticancer effects in prostate cancer when administered as standalone treatments [45,46,47,48,49,50,51,52], while Afatinib, Axitinib, Tranylcypromine, and Vorinostat were used in combination with other drugs or in a modified form in prostate cancer [53,54,55,56,57,58,59].

Of special interest, PDGFB (platelet-derived growth factor) plays a pivotal role in tumor angiogenesis and growth, facilitated by the consistent activation of platelets in the tumor microenvironment (TME), driven by comparable activation signals found in wound healing processes [60,61]. Our analysis further revealed that PDGFB is the target of several approved drugs including Sunitinib, Afatinib, Axitinib, and Imatinib. These drugs belong to the class of tyrosine kinase inhibitors and are widely utilized for the treatment of various cancer types (Table 2). While these drugs have been used to treat specific types of prostate cancer, there is a gap in research specifically addressing their efficacy in the context of *FOXA1* mutation. Furthermore, in prostate cancer, the increased expression of KDM1A is associated with cancer progression [62]. Consequently, there is a growing interest in its potential as a therapeutic target for cancer treatment. Extensive research into a range of KDM1A inhibitors is currently underway, with rigorous clinical evaluations, highlighting their potential as promising candidates for cancer therapy. In our study, we specifically identified Tranlcypromine, Phenelzine, Rosiglitazone, Pioglitazone, Pargyline, and Vorinostat as KDM1A target drugs. Notably, while Vorinostat was only approved for cutaneous T cell lymphoma, the other drugs were originally approved for depression, diabetes, or hypertension [63]. Another potential therapeutic gene we focused on was *MAOA* (monoamine oxidase A). MAOA is closely associated with the AR activity and development of prostate cancer [64,65]. Phenelzine, Selegiline, Pargyline, Moclobemide, Nomifensine, and Minaprine emerged as MAOA target drugs in this study. Supporting our result, previous studies examined the anticancer effects of MAO inhibitor. Pargyline and Phenelzine decreased growth and proliferation of androgen-sensitive and castration-resistant prostate cancer cells [51,66]. The first clinical trial of MAO inhibitor, Phenelzine, was performed, and lower prostate-specific antigen levels in more than half of the participants were observed. The anticancer effects of MAO inhibitors are not limited to prostate cancer [49]. Its effects were reported in glioma, breast cancer, and non-small cell lung cancer, indicating its broader anti-tumorigenic effects [67,68,69]. Considering previous studies and our results, we suggest that MAOA inhibitors might have a special efficacy on *FOXA1* mutant prostate cancer compared to the control group. Furthermore, HSP90AB1 emerged as another potential therapeutic target due to its direct interactions with FOXA1. Although HSP90AB1 was not included in the CancerMine prostate cancer genelist, HSP90 family proteins were previously known for their cancer proliferative activity in prostate cancer, such as HSP secretion by tumor which initiates EMT [70,71]. Notably, Dipyridamole and Diacerein were identified as potential drugs targeting HSP90AB1. Dipyridamole has previously been reported to exhibit antiproliferative activity in prostate cancer [52]. Furthermore, a study has shown that Diacerein mitigates benign prostatic hyperplasia induced by testosterone in rats [72]. However, there is limited information regarding the efficacy of Diacerein in prostate cancer.

In conclusion, we suggested that *FOXA1* mutations may also be useful as a predictive biomarker for response to certain therapies, highlighting the potential importance of precision medicine in the treatment of prostate cancer. By identifying patients with *FOXA1* mutations and tailoring their characteristics accordingly, we investigated which genes and pathways altered and how immune landscapes changed in *FOXA1* mutant prostate cancer. To increase reliability, we need further in vitro and in vivo studies as well as in silico studies with a larger number of *FOXA1* mutation samples. Nevertheless, our study provides a deeper understanding of the molecular mechanisms underlying a certain type of prostate cancer and promising therapeutic strategies, contributing to effective and personalized approaches to treatment that improve outcomes for patients.

## 4. Materials and Methods

### 4.1. Sample RNA-Seq Data Acquisition

We downloaded processed TCGA PRAD RNA-seq data with normalized genes and clinical data including pathological information from the Broad GDAC Firebrowse website (http://gdac.broadinstitute.org/, accessed on 13 January 2022) and cBioPortal for Cancer Genomics website (https://www.cbioportal.org/, accessed on 27 February 2022). Only primary solid tumor samples with the TCGA sample type code ‘-01’ were subtracted and used for the following analysis (https://docs.gdc.cancer.gov/Encyclopedia/pages/TCGA_Barcode/, accessed on 27 February 2022). Detailed information of *FOXA1* mutations was visualized and downloaded from cBioPortal.

### 4.2. Case and Control Sample Selection

*FOXA1* mutation samples (*n* = 9, mutant group) were selected based on the classification criteria of the TCGA molecular taxonomy study of primary prostate cancer [4]. The mutation-negative (control) group was selected from the ‘other’ subtype (*n* = 86) from the same paper. They have no other distinct features; *ERG* fusions, *ETV1/ETV4/FLI1* fusions, or overexpression; or *SPOP*, *FOXA1*, and *IDH1* mutations. To remove false-positive results due to the difference of sample number between *FOXA1* mutant and control groups, additional criteria were applied: (1) 67 patients in the control group with total mutation counts between 25 and 90 were only considered because total mutation counts in the *FOXA1* mutant group were between 29 and 89. (2) Among the 67 patients in the control group, 21 outliers in the expression levels of *FOXA1* and *AR* were excluded. Finally, 46 cases were set as the control group (Table 1).

### 4.3. Gene Selection through DEG Analysis

To identify differently expressed genes (DEGs) between the *FOXA1* mutation group and the control group, statistical analysis was performed using Student’s *t*-test. Genes with *p*-values of less than 0.05 were considered statistically significant. Some 1525 genes satisfied this criterion, so were considered as significant DEGs. The mean expression values of each selected gene were used to calculate fold changes (FCs) between the *FOXA1* mutation group and the control group. FC = (mean expression in *FOXA1* mutation group)/(mean expression in control group). Genes with FC > 1 were considered as upregulated genes, and genes with FC < 1 were considered as downregulated genes in the mutant group.

### 4.4. Pathway Analysis via ConsensusPathDB (CPDB)

Some 1525 genes from DEG analysis were applied as inputs for over-representation analysis (ORA) using ConsensusPathDB (CPDB, http://cpdb.molgen.mpg.de/, accessed on 7 May 2022) [73]. Pathway databases such as Wikipathways, SMPD, KEGG, Reactome, PharmGKB, PID, Biocarta, Ehmn, Humancyc, INOH, Netpath, and Signalink were included. Altered signaling pathways according to *FOXA1* mutation were selected with a minimum overlap input list (*n* = 2) and *p*-value cutoff (*p* < 0.05). Eight cancer-related pathways with 132 genes and six immune-associated pathways with 56 genes were selected.

### 4.5. Study Validation Using GEO Dataset

The results in the current study were validated using the GEO dataset series GSE123619. All RNA-seq samples were derived from the 22Rv1 cell lines, and the *FOXA1* mutation was induced using the CRISPR system [12]. GSM3508126 and GSM3508128 were used for control, and GSM3508127 and GSM3508129 were used for *FOXA1* mutation. The expression table was downloaded and fold changes as (mean expression in *FOXA1* mutation)/(mean expression in control) were calculated. Genes that met the condition of FC < 0.8 or FC > 1.3, resulting in a total of 2001 genes, were considered significant and subjected to over-representation analysis.

### 4.6. Tumor-Immune Phenotypes Profiling in FOXA1 Mutant Prostate Cancer

TIMER (http://timer.cistrome.org/, accessed on 20 July 2022) is a systematical bioinformatic tool designed for analyzing immune infiltration across various cancer types [34,74]. TIP (Tracking Tumor Immunophenotype, http://biocc.hrbmu.edu.cn/TIP/index.jsp, accessed on 20 July 2022), another practical tool for immune phenotype analysis, portrays the status of anticancer immunity throughout a seven-step Cancer-Immunity Cycle encompassing release of cancer cell antigens (Step 1), cancer antigen presentation (Step 2), priming and activation (Step 3), trafficking of immune cells to tumors (Step 4), infiltration of immune cells into tumors (Step 5), recognition of cancer cells by T cells (Step 6), and killing of cancer cells (Step 7) [22]. The above analytic tools were used to provide immune phenotypic results for the selected TCGA data.

### 4.7. Selection of Significant Cancer-Related Genes and Potential Drugs

We obtained a list of cancer genes, including cancer driver genes, oncogenes, and tumor suppressor genes specific to prostate cancer, from the CancerMine database (http://bionlp.bcgsc.ca/cancermine/, accessed on 22 September 2022) [75]. These genes were further annotated with the DEGs identified in relation to *FOXA1* mutation, in order to select *FOXA1*-associated prostate cancer specific-cancer genes. Actionable drugs that target *FOXA1*-associated prostate cancer specific-cancer genes were searched via CADDIE (https://exbio.wzw.tum.de/caddie/, accessed on 22 September 2022), a platform integrating multiple drug databases, including BioGRID, DrugBank, ChEMBL, and DGIdb [76]. Protein–protein interactions were investigated and visualized via STRING [23].

### 4.8. Data Visualization

Software R version 4.1.3 was used for statical computing and graphics. Patient characteristics in Table 1 were created using the moonBook package in R. DEG results were visualized by the Complexheatmap, Circlize, and EnhancedVolcano packages in R. GraphPad Prism version 9.4.1 was used to create the dot graph illustrating eight cancer-related pathways (Figure 2A), the comparison between the TCGA and GEO datasets (Figure 2D and Figure 3D), and the survival plot (Appendix A). The circular plot showing overlapped genes in pathway analysis and the gene enrichment graph (Figure 2C and Figure 3A) were constructed by SRplot (http://www.bioinformatics.com.cn/srplot, accessed on 13 February 2023). AR-related pathways and B-cell-related pathways (Appendix A) were further investigated with GSEA 4.3.2 (https://www.gsea-msigdb.org/gsea/index.jsp, accessed on 15 May 2023) [77,78].

## Figures and Tables

**Figure 1 ijms-24-15823-f001:**
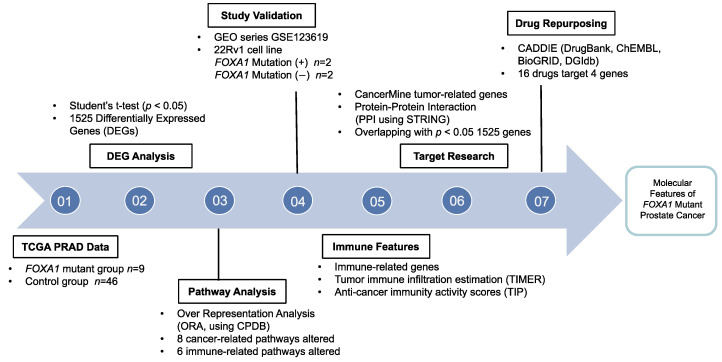
Study flowchart. This analytic study explores key features of *FOXA1* mutant prostate cancer using various bioinformatic methods and tools.

**Figure 2 ijms-24-15823-f002:**
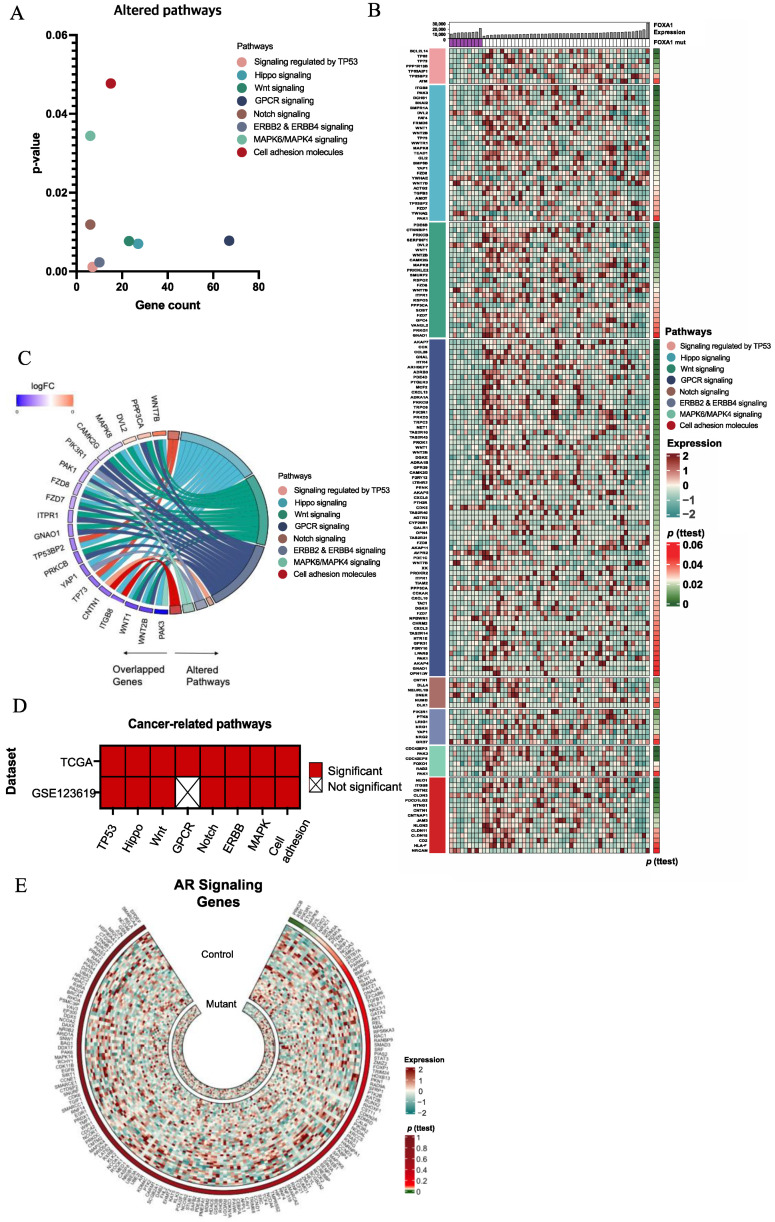
Cellular signaling pathways associated with *FOXA1* mutant prostate cancer. (**A**) Eight significant cancer-related pathways derived from over-representation analysis (ORA). The *x*-axis represents the number of genes within the pathways and the *y*-axis represents *p*-values in the pathways. (**B**) A heatmap shows genes within the eight cancer-related pathways. (**C**) A circular plot shows shared genes within the eight altered cancer-related pathways. Fold change (FC) values indicate the average mRNA expression in the mutant group compared to the control group. Genes upregulated in the *FOXA1* mutant group are in red, while those upregulated in the control group are in blue. Line connections indicate genes shared among multiple pathways, with distinct colors representing different signaling pathways. (**D**) Eight cancer-related pathways in (**A**–**C**) are validated using GEO dataset (GSE123619). Some 2001 genes used for this analysis were selected based on the criteria of FC (mutation/control) < 0.8 or FC > 1.3. (**E**) A circular heatmap represents the expression of AR signaling-related genes in the *FOXA1* mutant group and control group.

**Figure 3 ijms-24-15823-f003:**
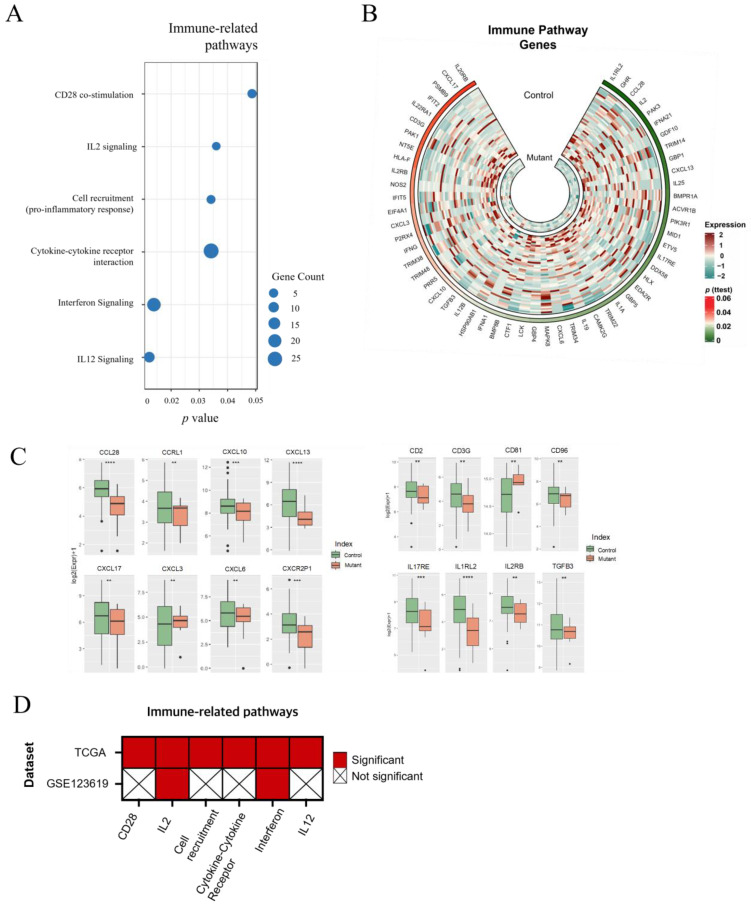
Immune-related gene and pathway analysis. (**A**) Six immune-related pathways are significantly affected by *FOXA1* mutation. Dot size indicates the number of genes in each pathway. (**B**) The expression of genes in the six immune-related pathways in *FOXA1* mutant and control groups are displayed by a circular heatmap. (**C**) The expression of various chemokines and cytokines genes in *FOXA1* mutant and control groups are plotted. (**D**) Six immune-related pathways in (**A**) are validated using GEO dataset (GSE123619). **: 0.01 ≤ *p* < 0.05, ***: 0.005 ≤ *p* < 0.01, ****: *p* < 0.005.

**Figure 4 ijms-24-15823-f004:**
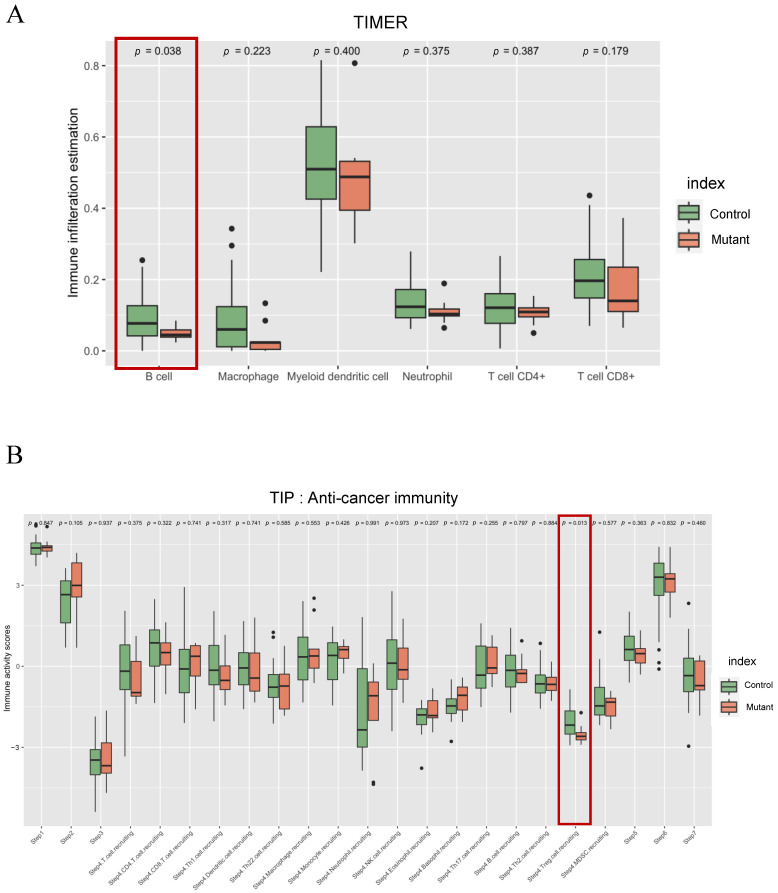
Immune-related tumor microenvironmental feature analysis. (**A**) Tumor-infiltrated immune cells are estimated in *FOXA1* mutant and control groups using TIMER 2.0. (**B**) The plot displays the 23 activity scores of anticancer immunities across seven steps of the cancer–immunity cycles between two groups. The scores are estimated using the tool TIP. The items marked in red boxes shows significant *p* values.

**Figure 5 ijms-24-15823-f005:**
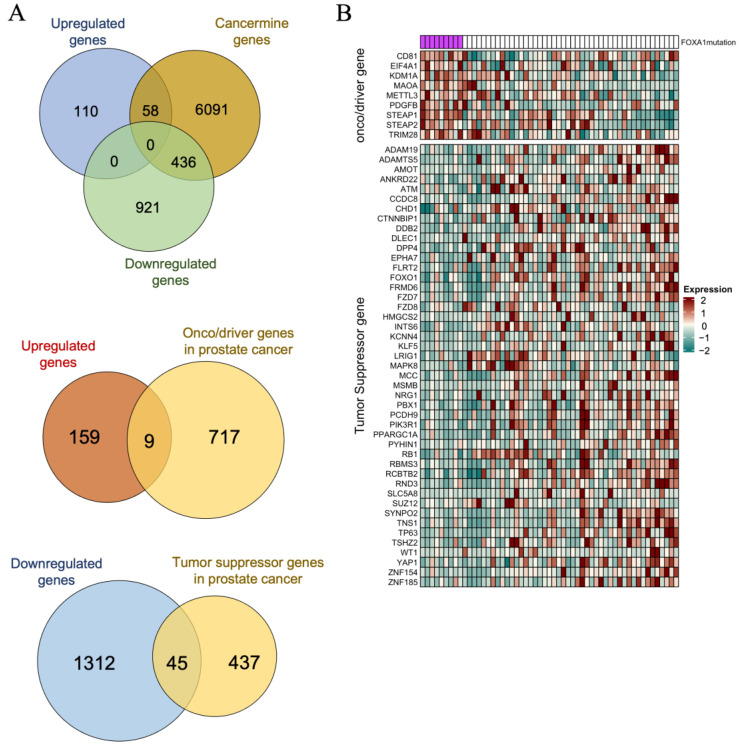
Selection of tumor-related significant genes. (**A**) Venn diagrams depict the number of genes overlapping with oncogenes, cancer driver genes, and tumor suppressor genes from CancerMine. Upregulated and downregulated genes were indicated by FC (mutation/control). Specifically, 168 upregulated genes overlap with 726 oncogenes/driver genes of prostate cancer, and 1356 downregulated genes overlap with 482 tumor suppressor genes. This analysis results in the discovery of 9 upregulated oncogenes/driver genes and 45 downregulated tumor suppressor genes in *FOXA1* mutant prostate cancer. (**B**) The expression of 9 upregulated oncogenes/driver genes and 45 downregulated tumor suppressor genes are shown in the heatmap.

**Table 1 ijms-24-15823-t001:** Patient characteristics. Patient characteristics of *FOXA1* mutant and control (*FOXA1* mutant-negative) group. Patients in each group are selected based on criteria mentioned in the method.

	*FOXA1* Mutant Group (*n* = 9)	Control Group (*n* = 46)	*p*
Age	65.3 ± 7.9	62.2 ± 5.9	0.241
Race			0.717
- Asian	0 (0.0%)	1 (4.0%)	
- Black or African American	0 (0.0%)	2 (8.0%)	
- White	5 (100.0%)	22 (88.0%)	
PSA Level	9.5 ± 5.2	11.1 ± 12.0	0.768
Reviewed Gleason Sum			0.046
- 6	0 (0.0%)	13 (28.3%)	
- 7	4 (44.4%)	25 (54.3%)	
- 8	4 (44.4%)	4 (8.7%)	
- 9	1 (11.1%)	3 (6.5%)	
- 10	0 (0.0%)	1 (2.2%)	
Primary Tumor Laterality			0.669
- Bilateral	8 (88.9%)	39 (86.7%)	
- Left	0 (0.0%)	3 (6.7%)	
- Right	1 (11.1%)	3 (6.7%)	
Mutation Count	47.9 ± 19.6	36.9 ± 8.5	0.134
Overall Survival (Months)	31.5 ± 26.5	37.1 ± 22.2	0.504
*AR* Expression	995.8 ± 744.8	866.6 ± 614.7	0.580
*FOXA1* Expression	13,196.3 ± 3475.4	11,111.0 ± 4343.1	0.181

**Table 2 ijms-24-15823-t002:** Summary of remarkable drugs targeting PDGFB, KDM1A, MAOA, and HSP90AB1. It illustrates a summary of drugs targeting PDGFB, KDM1A, MAOA, and HSP90AB1, along with their mode of actions (MOAs) and approved disease for which they are used.

Gene	Drug	MOA	Approved Diseases
PDGFB	Sunitinib *	an indolinone derivative and tyrosine kinase inhibitor with potential antineoplastic activity	gastrointestinal stromal tumor pancreatic caner renal cell carcinoma
	Afatinib ^†^	a tyrosine kinase receptor inhibitor that is used in the therapy of selected forms of metastatic non-small cell lung cancer	non-small lung cancer
	Axitinib ^†^	an orally bioavailable tyrosine kinase inhibitor; inhibits VEGF and PDGF, exerting an anti-angiogenic effect	renal cell carcinoma
	Imatinib	a protein–tyrosine kinase inhibitor that inhibits the BCR-ABL tyrosine kinase, PDGF and SCF, c-Kit, and inhibits PDGF- and SCF-mediated cellular events	acute lymphoblastic leukemia rare gastrointestinal cancer dermatofibrosarcoma protuberans (DFSP) myelodysplastic/myeloproliferative diseases (MDS/MPD) aggressive systemic mastocytosis (ASM)
KDM1A	Tranylcypromine ^†^	non-selective, irreversible monoamine oxidase inhibitor	major depressive disorder (MDD)
	Phenelzine *	non-selective, irreversible monoamine oxidase inhibitor	treatment-resistant depression, panic disorder, and social anxiety disorder
	Rosiglitazone *	insulin sensitizer, binding to the PPAR-r regulates the transcription of insulin-responsive genes	type-2 diabetes mellitus
	Pioglitazone *	stimulates PPAR-r enhances tissue sensitivity to insulin reduces the hepatic production of glucose	type-2 diabetes mellitus
	Pargyline *	monoamine oxidase B inhibitor	hypertension
	Vorinostat ^†^	histone deacetylase inhibitor inhibits HDAC1, HDAC2, and HDAC3 (Class I), HDAC6 (Class II)	cutaneous manifestations of cutaneous T-cell lymphoma (CTCL)
MAOA	Phenelzine *	non-selective, irreversible monoamine oxidase inhibitor	treatment-resistant depression, panic disorder, and social anxiety disorder
	Selegiline	selective, irreversible inhibitor monoamine oxidase B inhibitor	attention deficit hyperactivity disorder (ADHD) major depressive disorder (MDD) Parkinson’s Disease (PD)
	Pargyline *	selective, irreversible inhibitor monoamine oxidase B inhibitor	hypertension
	Moclobemide	reversible monoamine oxidase A inhibitor	major depressive disorder (MDD)
	Nomifensine	a norepinephrine-dopamine reuptake inhibitor increases the amount of synaptic dopamine available to receptors by blocking dopamine’s reuptake transporter	depression
	Minaprine	monoamine oxidase inhibitor binds to serotonin type-2 receptors and to dopamine D1 and D2 type receptors and serotonin reuptake pump	depression
HSP90AB1	Diacerein	a slow-acting medicine of the class anthraquinone inhibiting interleukin-1 beta	osteoarthritis
	Dipyridamole *	a nucleoside transport inhibitor a PDE3 inhibitor medication that inhibits blood clot formation	postoperative thromboembolic complications of cardiac valve replacement

*: Drugs reported to have anticancer effects in prostate cancer when treated alone. ^†^: Drugs reported to have anticancer effects in prostate cancer when treated with other drugs or in a modified form.

## Data Availability

Not applicable.

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
