# Peer review of "Unveiling the Molecular Landscape of FOXA1 Mutant Prostate Cancer: Insights and Prospects for Targeted Therapeutic Strategies"

_ijms, 2023, doi:10.3390/ijms242115823_

Round 1

Reviewer 1 Report

The manuscript, entitle, Unveiling the Molecular Landscape of FOXA1 Mutant Prostate 2 Cancer: Insights and Prospects for Targeted Therapeutic Strategies, is a detailed analysis of genome databases aimed at identifying differences between FOXA1 mutation status and gene expression profiles in prostate cancer tumor samples submitted for pathological information. Normalized gene and clinical data from the Broad GDAC and cBioPortal for Cancer Genomics websites were collected. Processed TCGA PRAD RNA-seq data was downloaded by the authors.

The initial analysis revealed that only 9 samples would be classified as having FOXA1 mutations, compared to 86 non-mutated. This points to the reality that although  FOXA1 ranks as one of the frequently mutated genes in prostate cancer, the frequency is low, reflecting the heterogeneity of the disease where p53, PTEN and other mutations are also important drivers. It also points to the difficulty in identifying clinically relevant mutations in a tumor where different areas may have different gene expression profiles. In addition, the authors do not discuss the location of mutations, the overall effect on function/activity, and the epigenetic status, which can greatly affect FOXA1 activity, maybe more than expression level.

As the authors point out, targeting FOXA1 has been challenging because of its function in non-malignant cells. Their plan to further analyze the data to find downstream pathways known to promote cancer progression to find genes that could be targeted with existing drugs has been used by others to great advantage. FOXA1 is known to be important in AR signaling and this may yet be important although the authors did not find a difference in AR gene signaling.

For validation, the authors  compared their findings with the GEO database, although, as they point out, there are limitations from using data from the 22Rv1 cells, known to have signaling abnormalities. This manuscript provides information that will be of interest to those embarking on in vitro and in vivo studies to better understand the progression of prostate cancer, interactions of tumor cells with immune infiltrating cells and other factors in the tumor  microenvironment. Investigating the use of existing agents, such as anti-angiogenic tyrosine kinase inhibitors, in combination and as single drug treatments will help improve outcomes for prostate cancer patients.

This manuscript is based entirely on analyses of information that was put into databases and depends on how well the predictive algorithms reflect actual biology, with respect to immune activation, microenvironment interactions and individual genetics, hormone levels, etc. As long as readers keep this in mind and use the information presented here as a basis for performing studies on patient tissue and cell lines, it will be of considerable value.

The quality of the English used is good with just a few minor typos/missing words.

Reviewer 2 Report

In their paper Won Hwang et al describe a cellular signaling pathways associated with FOXA1 mutant prostate cancer and specific genetic molecular changes from a biocomputational perspective. Although their findings may be of interest, a revision is required to meet the standards of the journal.

More specifically, the paper has both minor and major issues.

Minor issues:

-Line 100: In agreement with the rest of the text, please write 'nineteen' in number (19) instead of in letters.

-In Figure 2, the font size is too small, especially in paragraphs a, b, c, and e. Please enlarge the size or highlight the most interesting candidates above the rest.

-In Figure 5, the font size is too small, especially in paragraph b. Please enlarge the size or highlight the most interesting candidates above the rest.

-Figure 4a. In this figure, the authors demonstrate that B cells were significantly more abundant in the control group compared to FOXA1 mutant prostate cancer. However, this result is statistically significant only when using the TIMER tool, but not when utilizing MCP-counter, EPIC, xCell, quanTIseq, or CIBERSORT. Could the authors please provide an explanation for this discrepancy in the results? In this context, I recommend adding some clarification following lines 152-154,  "B cells were found to be significantly more abundant in the control group compared to FOXA1 mutant prostate cancer according to TIMER, but not according to other analysis tools.”

-Include Kaplan-Meier analysis to compare the survival of the mutant FOXA1 tumors group with respect to the control group.

Major issues:

-It is challenging to obtain robust conclusions solely through bioinformatic analyses without conducting experimental validations. In this regard, I believe that the sample size of mutant FOXA1 tumors (n=9) is too small. Could you please include additional studies to increase the sample size?

- Based on the alterations observed in Figures 3 and 4 regarding the immune system, further explanation, integration, and discussion of these results are required in the discussion section. A more in-depth discussion of these findings is needed

Reviewer 3 Report

Hwang et al. investigate the transcriptomic landscape of FOXA1-mutant prostate cancers, mining data from TCGA and performing RNAseq validation. The information is valuable and comes from a thoughtful hypothesis and detailed bioinformatic characterization of transcriptomics datasets.

The abstract is misleading. Firstly, the authors include strong statements about identifying novel therapeutic targets/drug candidates, when no experimental evidence is presented. Mere correlation from previous work is alluded to. The abstract must be revised to clearly illustrate where the authors derived this evidence from. “Furthermore, we identified novel therapeutic targets and potential drug candidates for FOXA1 mutant prostate cancer”….”Of particular interest, KDM1A, MAOA, and PDGFB emerged as promising druggable targets, benefiting from the existence of approved drugs. Notably, the effective targeting of MAOA and KDM1A using monoamine inhibitors presented a promising avenue specific to FOXA1 mutant prostate cancer.”

Secondly, the authors refer to the use of 22Rv1 cells for validation of TCGA data which is not an accurate reflection of the work performed. As written, it gives the impression that the authors cultured this cell line and performed RNAseq validation in the present study. But the authors essentially mined data from previous work that used the above cell line. The relevant work should be cited in the results section.  Although the original source of this dataset was mentioned in the discussion section, it is important to phrase this correctly in the abstract and results sections first.

Line 20 in Abstract. “…..which were significantly upregulated…”

Supplementary table 1: Please show genes enriched in each pathway from the dataset. Also show background genes.

Lines 91-93 in Section 2.2 of results: Please clearly mention here that differentially expressed genes were identified and used for pathway analysis.

Line 95 in Section 2.2: Include a volcano plot showing up and down-regulated genes (Log2) with Log10 p-value. Highlight genes of relevance on this plot.

Line 95 Section 2.2: The authors used pathway overrepresentation analysis. It therefore makes sense to identify separately pathways overrepresented from upregulated genes and downregulated genes. This will give at least some sense of pathway directionality.  

Line 376 in methods section: Did the authors use Log2-transformed FC? This needs to be clearly mentioned here.

Line 141: Rephrase “Considering these communications”

Line 112-114, 191-192: Grammatically incorrect sentences. Please rephrase appropriately.

Line 188-216: Section 2.5. “Identification of potential therapeutic targets and targeted drugs for FOXA1 mutant prostate 188 cancer patients” reads like a paragraph from a review article.

The drugs reported were largely non-selective and don’t significantly correlate with the target genes reported by the authors. At best- it seems that those drugs play an indirect role and therefore “might” show some effect. The authors are advised to identify drug linchpins by inferring druggable protein-interactions of the target genes. Use the dataset of differentially expressed genes and identify physical protein interactions among the overexpressed genes. Are there any interacting proteins which have selective drugs? Secondly, are any of the interacting proteins as well as the target proteins lethal to prostate cancer cell lines via DepMap database? Include a PPI network annotating druggable and lethal genes. The authors may either annotate selective drugs within this network or include that information in a supporting table.  

Language needs to be professionally revised while adhering to scientifically-accepted standards of stating results.

Round 2

Reviewer 2 Report

I appreciate the authors for taking into account my feedback.

Incorporating these revisions has enhanced the quality of the manuscript compared to its initial version.